# Field Investigation and Finite Element Analysis of Landslide-Triggering Factors of a Cut Slope Composed of Granite Residual Soil: A Case Study of Chongtou Town, Lishui City, China

**Tiesheng Yan [1], Jun Xiong [1], Longjian Ye [1], Jiajun Gao [2] and Hui Xu [2],***

1   Southern Zhejiang Comprehensive Engineering Surveying and Mapping Institute, Hangzhou 310030, China
2   School of Civil Engineering and Achitecture, Zhejiang Sci-Tech University, Hangzhou 310018, China
*   Correspondence: xuhui@zstu.edu.cn

**Abstract:** Landslides caused by excavations and precipitation events are widespread types of slope failures in southwest Zhejiang, China, in areas with granite residual soil. Investigations of the effect of high precipitation on the hydrological response, stability, and evolutionary mechanism of cut slopes in granite soil areas are lacking. The characteristics of historical landslides in Chongtou Town in southwestern Zhejiang were summarized, and a typical slope was selected for analysis. The hydraulic and mechanical properties of the residual soil and fully weathered granite were tested, and the surface displacements on the slope were monitored. Geo-studio was utilized to establish a coupled seepage-deformation model to validate the numerical method and investigate the landslide-triggering factors of the cut slope. The results showed nearly all historical landslides in Chongtou Town were triggered by precipitation events, and the slide bodies consisted of residual soil and fully weathered granite with similar geotechnical properties. The simulated and measured horizontal displacements were in good agreement, indicating the reliability of the established model and parameters. The stability coefficient decreased with an increase in the gradient or height of the cut slope. The critical height values were 5.3 m, 5.5 m, 5.7 m, 6.0 m, and 6.3 m at slopes of 60°, 65°, 70°, 75°, and 80°, respectively. Long-term torrential rain and short-term high-intensity precipitation events are likely to trigger landslides when the precipitation event lasts longer than 26 h and 78 h, respectively. The landslide formation includes four stages: slope evolution, formation of unloading zone at slope foot, migration and loss of soil particles, and instability of the cut slope. The findings can be used to prevent and manage landslides on cut slopes in areas with granite residual soil.

**Keywords:** landslide; granite residual soil; cut slope; engineering geological investigation; field displacement monitoring; laboratory geotechnical test; coupled seepage-deformation analysis

## 1. Introduction

Granite is widely distributed in southwest Zhejiang, China, and is affected by regional tectonic movement. Climatic conditions and long-term weathering produce granite residual soil and a weathered layer [1]. Due to an increase in infrastructure construction projects, residential areas are being built in areas with unfavorable geologic conditions. Many cut slopes have been created for construction in granite residual soils highly susceptible to landslides, which can cause significant property damage, injuries, and loss of life [2–4]. For example, on 28 September 2016, a landslide occurred in Su village in the municipality of Suichang County, Lishui. It caused 26 deaths, two people were lost, and the economic loss amounted to more than 50 million yuan [5]. An investigation of the landslides' cause showed that the granite in the area was strongly weathered, engineering activities had modified the slope, and the precipitation amount on the day of the landslide was 127.2 mm.

Many numerical and experimental studies have been conducted on the instability of cut slopes in recent decades. Tu et al. [6] and Xu et al. [7] found that steep cut slopes increased the shear stress concentration at the foot of the slope and tension stress at the top,

leading to internal crack formation. The water entered the cracks and penetrated the deeper sections of the slope, resulting in slope failure. Zhang and Wang [8] found that the critical conditions for the propagation of catastrophic shear zones on engineering slopes were related to the gravitational shear stress ratio of the slope, which is influenced by the slope toe. Qin et al. [9] observed that key sections of the slope affected its stability, and the slope toe was the dominant section. The underlying rock is exposed at the cut slope toe, and the rock strength decreases over time due to weathering [10]. Panthee [11] investigated the relationship between the slope stability coefficient and the structure of the cut slope. It was found that reducing the gradient significantly enhanced slope stability. A similar result was reported by He et al. [12]. Cao et al. [13] found that the intrinsic cause of landslides on cut slopes in loess areas was the structural characteristics of the small soil pores. Mei et al. [14] and Sun et al. [15] observed similar results. Postill et al. [16] investigated the effect of excess pore water pressure at the cut slope toe on slope stability; it was found that cutting the slope caused damage and provided favorable conditions for a landslide disaster. Wang et al. [17] analyzed a cut slope with weathered metamorphic rock in southern Jiangxi using a custom-designed rainfall simulation system. The deformation and failure processes of the steep cut slope under heavy precipitation were divided into three stages: slipping of the slope surface, formation of tensile cracks on the platform, and slope collapse. Jin et al. [18] used Geo-studio numerical simulation software to study the rainfall infiltration of high-cut slopes during different rainfall cycles. The results showed that the shape and position of the potential sliding surface did not change significantly during rainfall infiltration. The depth of rainfall infiltration and the degree of stability degradation of the high-cut slope were positively correlated with the rainfall intensity. Pradhan et al. [19] performed a detailed slope stability analysis of 20 vulnerable cut slopes ranging from Rishikesh to Devprayag in the Himalayas using the Phase 2D finite element simulator. It was found that the nonlinear generalized Hoek-Brown (GHB) criterion provided better results than the Mohr–Coulomb (MC) criterion for the jointed rock common in the Himalayas. Suggestions were provided to strengthen the stability of cut slopes. Luo et al. [20] used a multi-wedge translation mechanism to analyze the stability of high-cut slopes reinforced with piles. The results indicated that the safety factor of the slope depended on the excavation depth. The slope stability was influenced by the position of the reinforced piles, the excavation distance, and the slope angle. The most suitable position of the reinforced piles was determined. Katz et al. [21] used numerical modeling to assess rock fall hazards and associated risks in the Soreq and Refaim valleys. They showed that rock falls caused by earthquakes damaged the road network. Chirico et al. [22] examined the role of vegetation on slope stability in the unsaturated region of shallow soils. It was found that vegetation minimized soil loss and stabilized cut slopes because the vegetation-soil system enhanced the soil's shear strength [23,24]. However, recent studies have shown that vegetation degradation and soil erosion may occur several years after slope revegetation [25–27]. Huang et al. [28] developed a high-cut slope risk evaluation model using a backpropagation (BP) neural network algorithm. The results showed that the risk evaluation level was II. The primary risk factors were earth excavation, scaffolding equipment, slope height, slope rate, groundwater level, personnel safety awareness, and construction safety risk management.

Previous studies have obtained many valuable findings, contributing to an in-depth understanding of landslide-triggering factors of cut slopes. However, most studies focused on the instability of cut slopes on the underlying rock or in loess areas. In addition, only individual landslides were investigated in most cases. Therefore, this study considers data on historical landslides and investigates a cut slope composed of granite residual soil in southwest Zhejiang Province. Laboratory tests are carried out to analyze the mechanical and hydraulic characteristics of typical soil samples. The effects of the cut slope's height and gradient and the precipitation intensity on the seepage stability of the slope are quantitatively investigated using Geo Studio software. The results can be used to prevent landslide disasters in granite soil areas in southwest Zhejiang, China.

## 2. Study Area

### 2.1. Regional Geological Setting

Chongtou Town is located in the southwest of Yunhe County, Lishui City, Zhejiang Province (119°20′47″ E–119°31′23″ E, 27°53′35″ N–28°6′44″ N) and covers an area of 228.2 km². As shown in Figure 1, acidic intrusive rocks, mainly granite, are common north of Chongtou Town, covering an area of 49.3 km² and accounting for 21.6% of the total town area. The road network of the village roads has high connectivity. The population in the villages accounts for 61.3% of Chongtou Town.

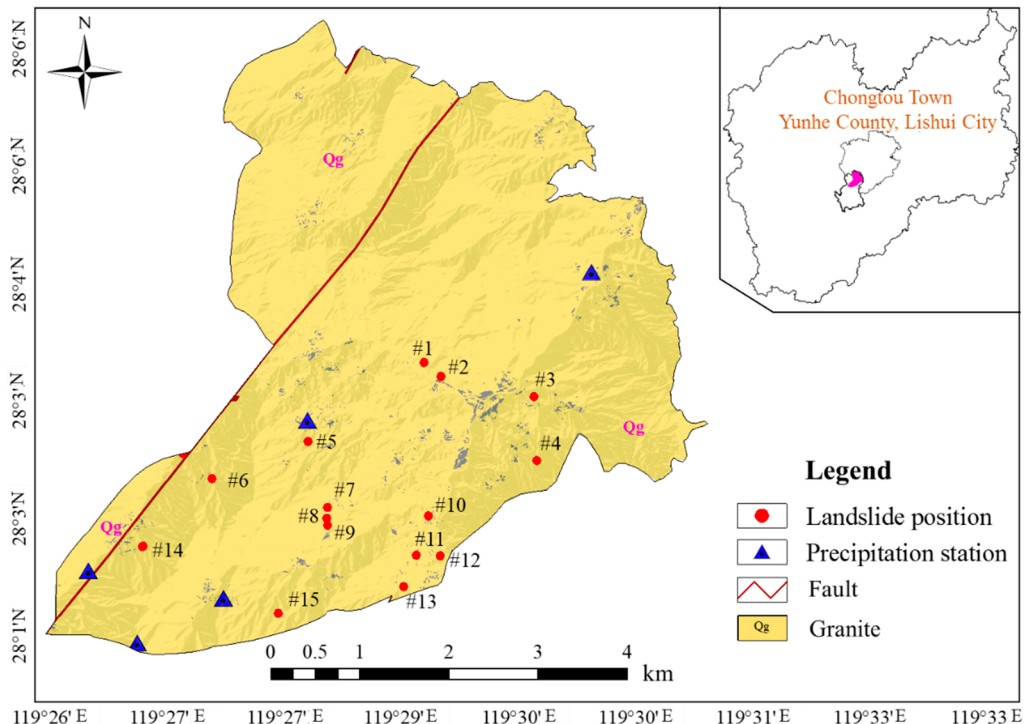

**Figure 1.** Petrographic diagram of the study area.

### 2.2. Characteristics of Historical Landslides

The data on historical landslides in granite soil areas of Chongtou Town obtained from high-precision geological hazard surveys are listed in Table 1. Fifteen landslide events have been recorded since 1997 (Figure 1). They occurred at the foot of cut slopes (an example is shown in Figure 2a,b), with volumes ranging from 200 m³ to 9500 m³. The slope height where landslides have occurred is 58–116 m, and the slope gradient is 15–35°. The height of the cut slope at the toe is 2–10 m, and the slope gradient is 60–80°. Figure 2c shows soil samples obtained from drilling. The thicknesses of the residual soil and fully weathered granite layer are 2–5 m and 15–30 m, respectively. Figure 3 shows the cross-sections of selected historical landslides. The precipitation data related to the landslide occurrence were obtained from the county meteorological station and analyzed statistically (Figure 4). The precipitation levels, according to the China Meteorological Administration, are 35 mm/d (general), 75 mm/d (heavy), 150 mm/d (torrential), and 200 mm/d (extreme). Precipitation events with different intensities and durations occurred within 168 h before the landslide. Most intensity levels were general or heavy, but the precipitation intensity was the highest on the date of the landslide (heavy or torrential).

**Table 1.** Statistics of historical landslides.

| Number | Date | Landslide Volume ($m^3$) | Height of the Natural Slope ($h$) (m) | Gradient of the Natural Slope ($\alpha$) (°) | Height of the Cut Slope ($h$) (m) | Gradient of the Cut Slope ($\theta$) (°) | Thickness of Residual Soil (m) | Thickness of Fully Weathered Granite (m) |
|---|---|---|---|---|---|---|---|---|
| #1 | 1 August 2011 | 300 | 65 | 15~25 | 2~8 | 65~70 | 2~3 | 20~25 |
| #2 | 1 June 2014 | 2400 | 83 | 15~25 | 4~8 | 60~70 | 3~4 | 20~25 |
| #3 | 14 June 2014 | 9500 | 96 | 25~35 | 4~8 | 65~80 | 3~5 | 25~30 |
| #4 | 6 July 2010 | 200 | 112 | 25~35 | 6~8 | 65~70 | 2~3 | 15~20 |
| #5 | 13 September 2015 | 1200 | 79 | 25~35 | 3~10 | 60~70 | 2~3 | 20~25 |
| #6 | 9 July 2020 | 1350 | 91 | 25~35 | 6~10 | 60~75 | 3~4 | 20~25 |
| #7 | 1 May 2015 | 800 | 116 | 25~35 | 5~8 | 65~80 | 2~3 | 20~25 |
| #8 | 4 July 1997 | 7400 | 89 | 15~25 | 5~10 | 65~80 | 3~5 | 25~30 |
| #9 | 8 July 2020 | 320 | 105 | 25~35 | 2~6 | 65~75 | 2~3 | 15~20 |
| #10 | 11 October 2008 | 450 | 58 | 15~25 | 2~6 | 60~75 | 2~3 | 15~20 |
| #11 | 21 September 2015 | 500 | 95 | 25~35 | 5~10 | 65~80 | 2~3 | 15~20 |
| #12 | 17 April 2017 | 300 | 82 | 15~25 | 2~8 | 65~70 | 2~3 | 15~20 |
| #13 | 1 August 2009 | 1350 | 61 | 25~35 | 2~8 | 60~70 | 3~4 | 20~25 |
| #14 | 23 July 2010 | 800 | 73 | 15~25 | 2~8 | 60~75 | 2~3 | 20~25 |
| #15 | 7 October 2013 | 4400 | 69 | 15~25 | 3~9 | 65~80 | 3~5 | 25~30 |

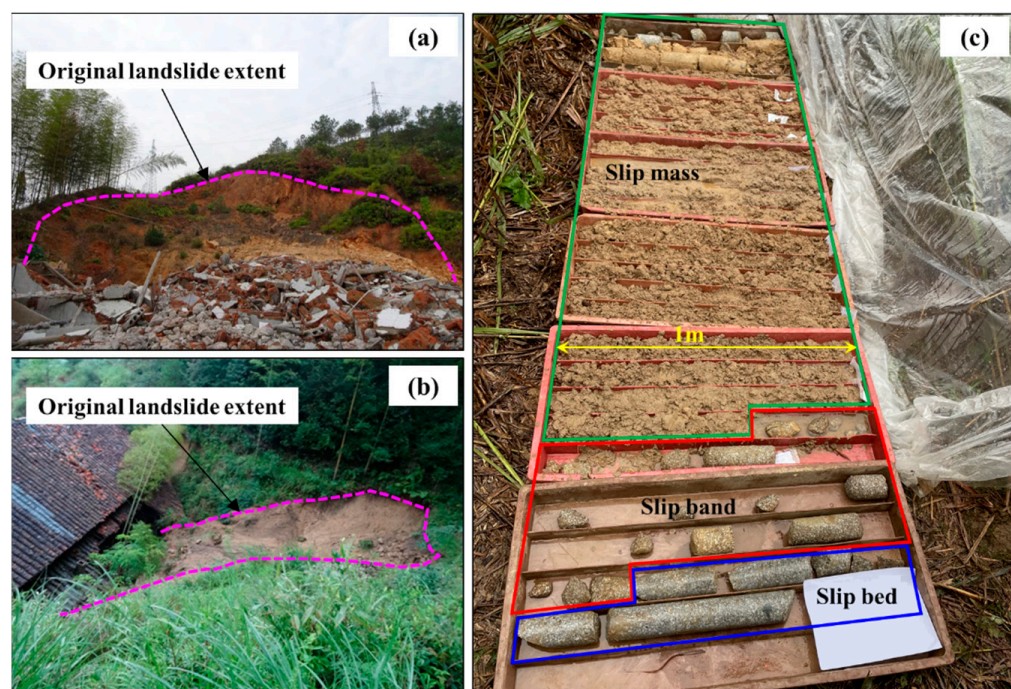

**Figure 2.** Field photographs of historical landslides: (**a**,**b**) Landslide patterns; (**c**) Soil samples obtained from drilling.

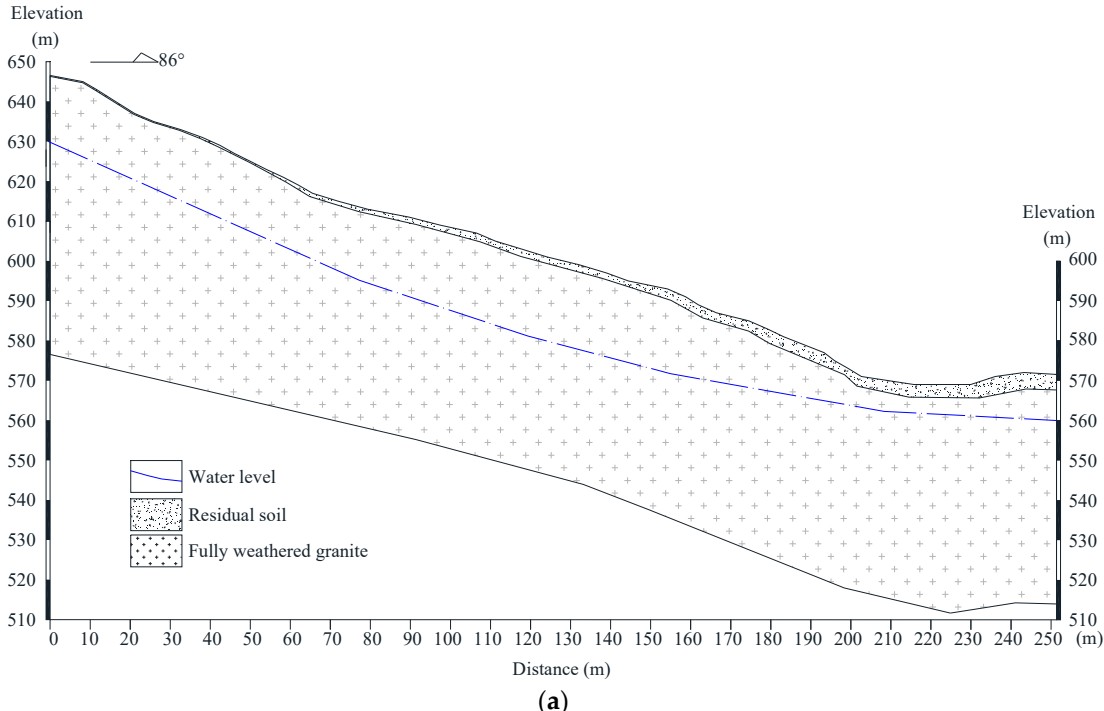

(**a**)

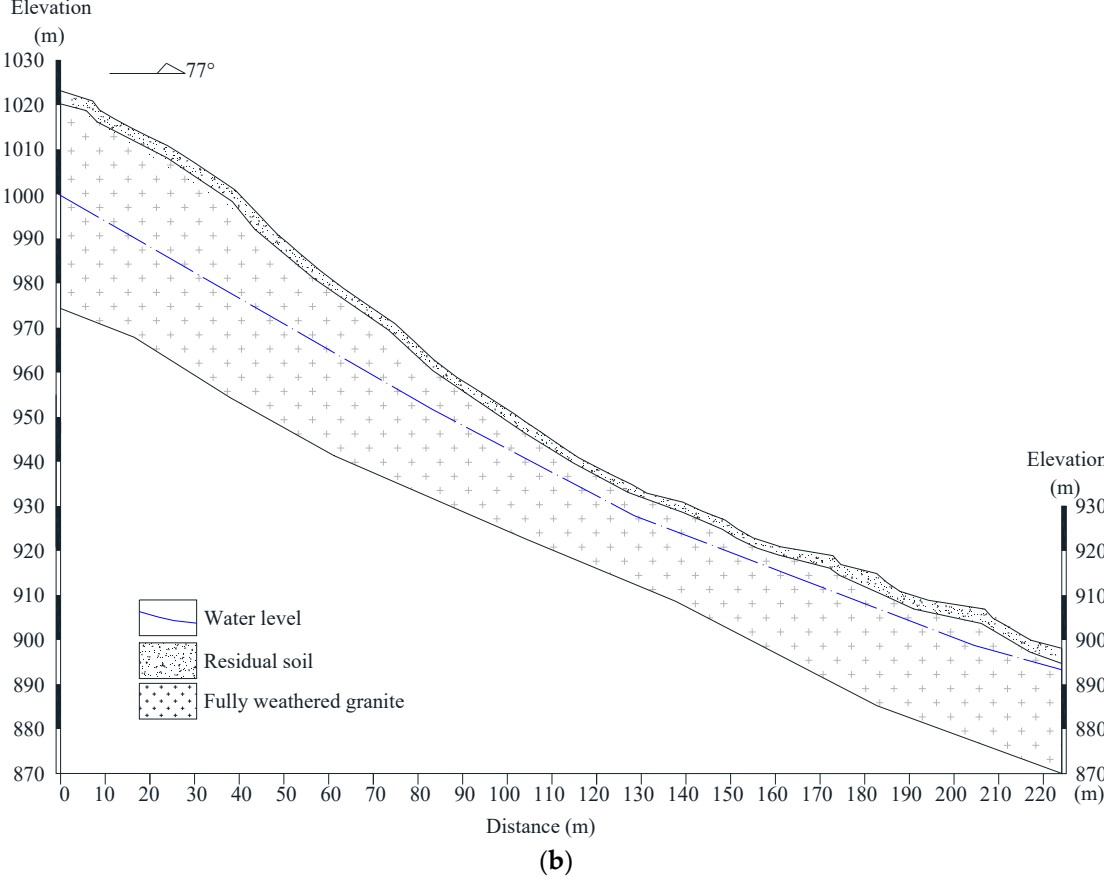

(**b**)

**Figure 3.** *Cont.*

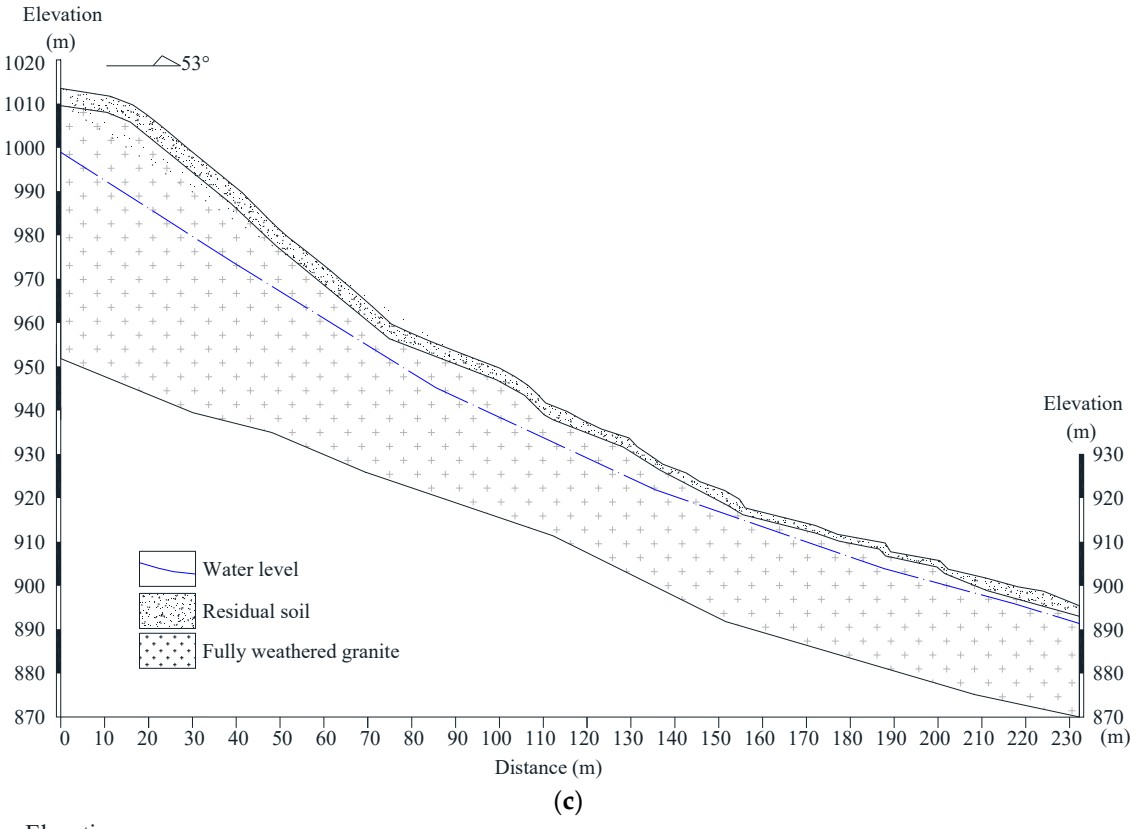

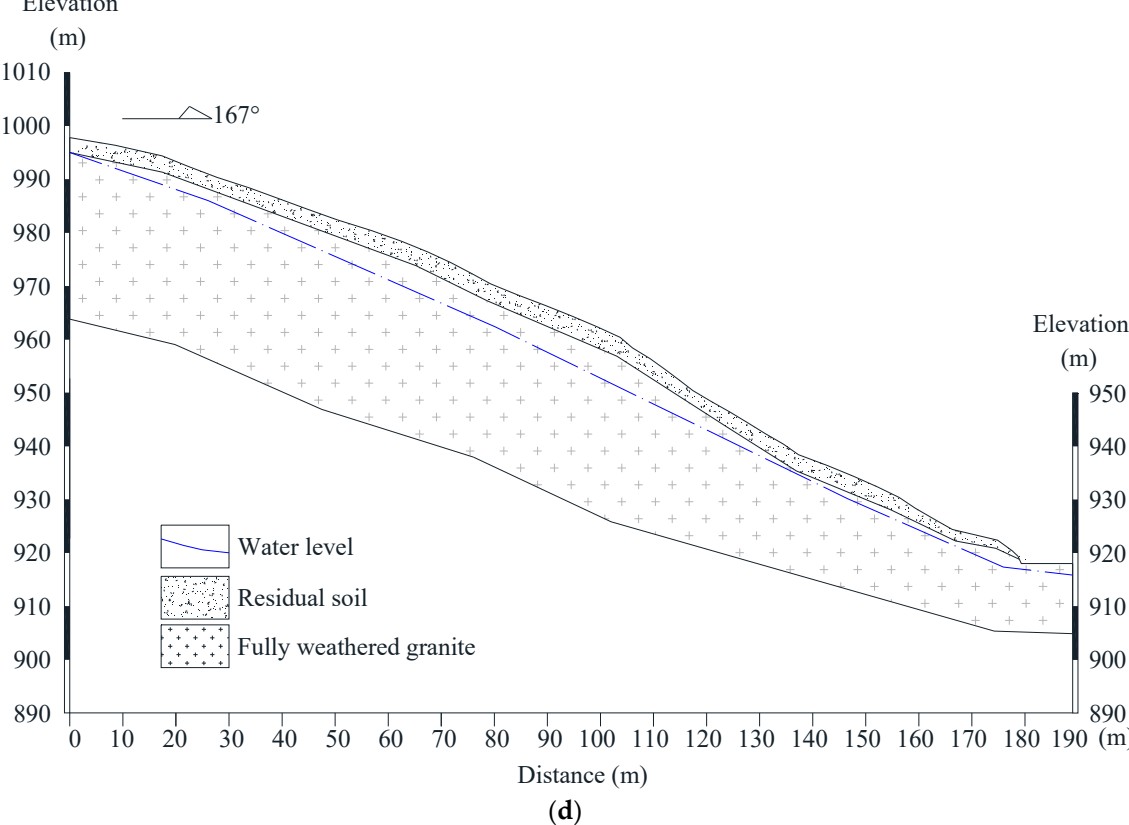

**Figure 3.** Cross-section diagrams of selected historical landslides. (**a**) #2 landslide; (**b**) #5 landslide; (**c**) #8 landslide; (**d**) #13 landslide.

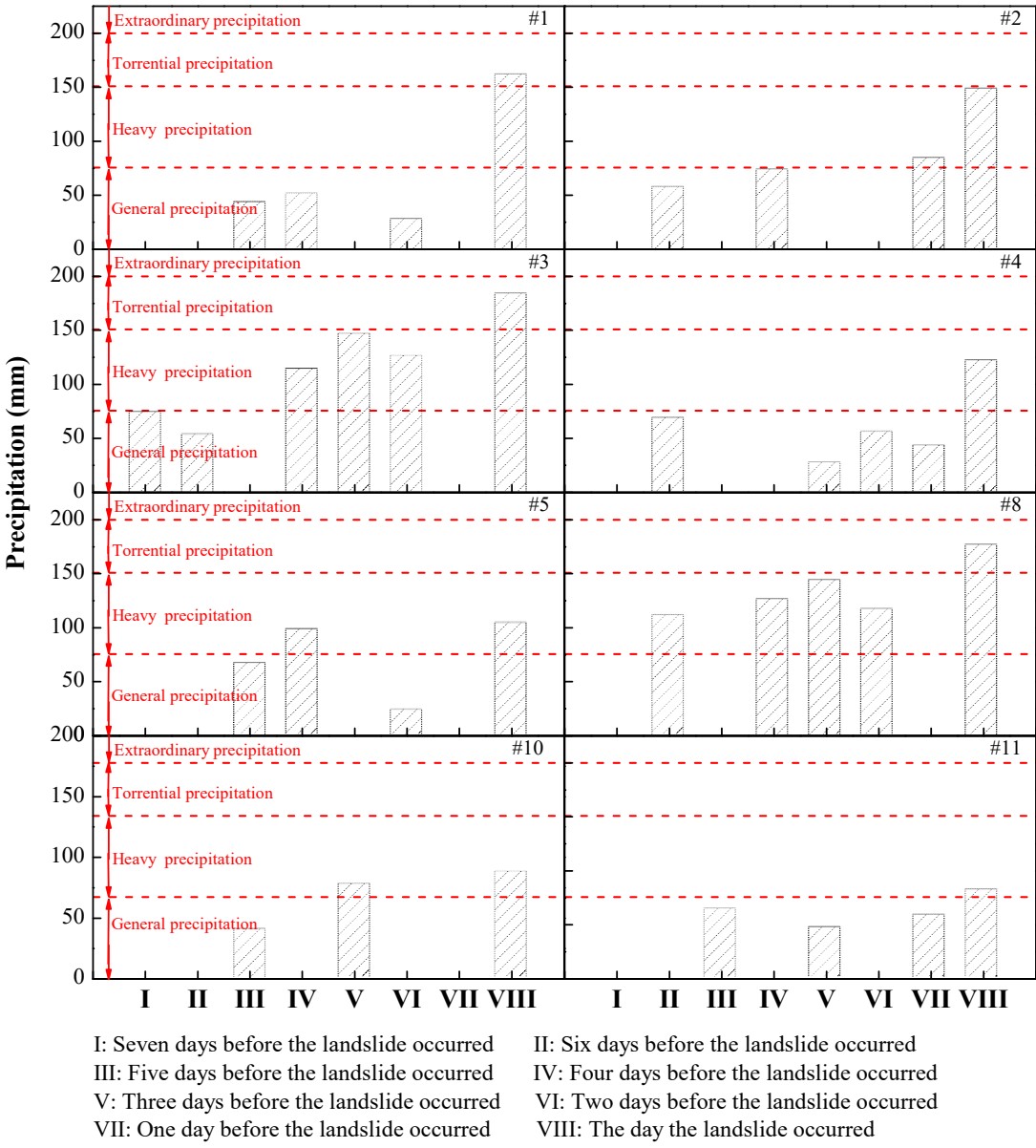

**Figure 4.** Precipitation data related to historical landslides.

*2.3. Engineering Conditions of a Typical Slope*

The selected slope is located in Xiayang Village, Chongtou Town. The area has many cut slopes created by excavating the foot of the slope for renovating old houses. Since late June 2022, small landslides have repeatedly occurred on cut slopes behind villagers' houses. After the events were reported, the local government and departments immediately dispatched technicians to investigate the slope on 3 July. As shown in Figure 5, the investigation results show many undesirable geological phenomena from the slope's foot to the central area, where human activities are frequent, such as cracks in dirt roads and small slides at the edge of farmland. From 9–14 July, the township government evacuated some of the residents, and the survey department conducted geological drilling on the slope. Figure 6 shows the cross-section diagram. Monitoring devices (YT-DG-0705, YiTuo sensing technology, Changsha, China) to record horizontal surface displacement were installed at the top of the cut slope, and three displacement monitoring points (M1, M2, and M3) were established. The parameters of the typical slope are listed in Table 2.

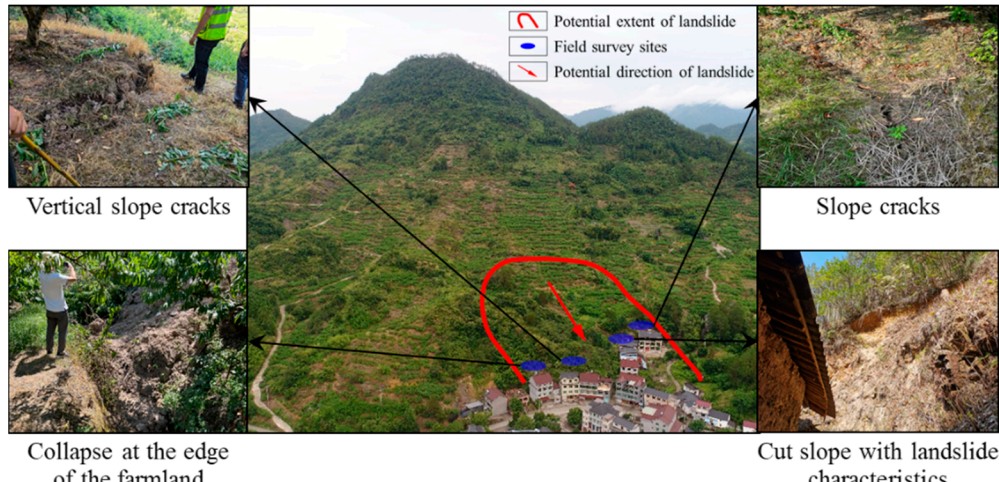

**Figure 5.** Photographs of the typical slope.

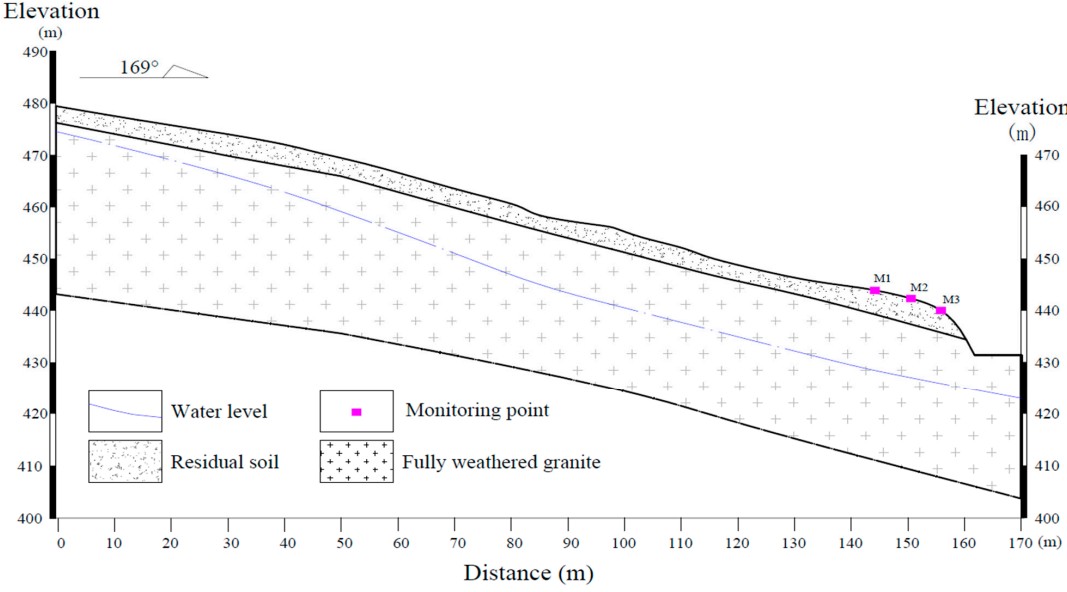

**Figure 6.** Cross-section diagram of the typical slope.

**Table 2.** Parameters of the typical slope.

| Parameters | Value |
|---|---|
| Height of the natural slope (*H*) (m) | 75 |
| Gradient of the natural slope (*α*) (°) | 27 |
| Height of the cut slope (*h*) (m) | 8 |
| Gradient of the cut slope (*θ*) (°) | 65 |
| Thickness of residual soil (m) | 2.8 |
| Thickness of fully weathered granite (m) | 30 |

## 3. Materials and Methods

### 3.1. Laboratory Test

Since the residual soil layer is thin, and the fully weathered granite layer is thick, samples of the residual soil were collected at different positions, and the fully weathered soil was sampled at different depths during drilling. The physical and mechanical parameters

of the soil were analyzed, including the initial dry density, weight, shear strength, saturated permeability coefficient, Poisson's ratio, and deformation modulus. All test methods followed the "Geotechnical test method standard (GB/T 50123-2019)" [29]. The results (Table 3) of the parameters are the average of the sample values.

**Table 3.** Physical properties of residual soil and fully weathered granite.

| Soil Layer | Initial Dry Density (g/cm³) | Unit Weight (kN/m³) | Cohesion (kPa) | Internal Friction Angle (°) | Saturated Permeability Coefficient (m/s) | Poisson's Ratio | Deformation Modulus (MPa) |
|---|---|---|---|---|---|---|---|
| Residual soil | 1.34 | 18.6 | 14.3 | 12.1 | $5.3 \times 10^{-5}$ | 0.32 | 13.7 |
| Fully weathered granite | 1.42 | 19.6 | 15.4 | 18.7 | $5.6 \times 10^{-5}$ | 0.28 | 31.5 |

A pressure film instrument (Cat. No. 1600, SEC Corporation, Washington, DC, USA) was used to test the soil and derive the soil water characteristic curves (SWCCs) of the residual soil and fully weathered granite (Figure 7). The van Genuchten-Mualem (VG-M) model [30] was used to fit the experimental data points to obtain the SWCC (Equation (1)). The fitting parameters are listed in Table 4. Equation (2) was used to fit the permeability coefficients of the SWCC of the residual soil and fully weathered granite. The results are shown in Figure 7.

$$\theta = \theta_{\mathrm{r}} + \frac{\theta_{\mathrm{s}} - \theta_{\mathrm{r}}}{\left(1 + (\alpha|h|)^{n}\right)^{m}} \tag{1}$$

where $\theta$ is the volumetric water content; $h$ is the matric suction, which was obtained from the laboratory test; $\theta_{\mathrm{r}}$ is the residual volumetric water content; $\theta_{\mathrm{s}}$ is the saturated volumetric water content; $m$, $n$, and $\alpha$ are fitting parameters, with $m = 1 - 1/n$.

$$k = k_{\mathrm{s}} \frac{\left(1 - (\alpha|h|)^{(n-1)}\left(1 + (\alpha|h|)^{n}\right)^{-m}\right)^{2}}{\left(1 + (\alpha|h|)^{n}\right)^{m/2}} \tag{2}$$

where $k$ and $k_{\mathrm{s}}$ are the unsaturated and fully saturated permeability coefficients, respectively.

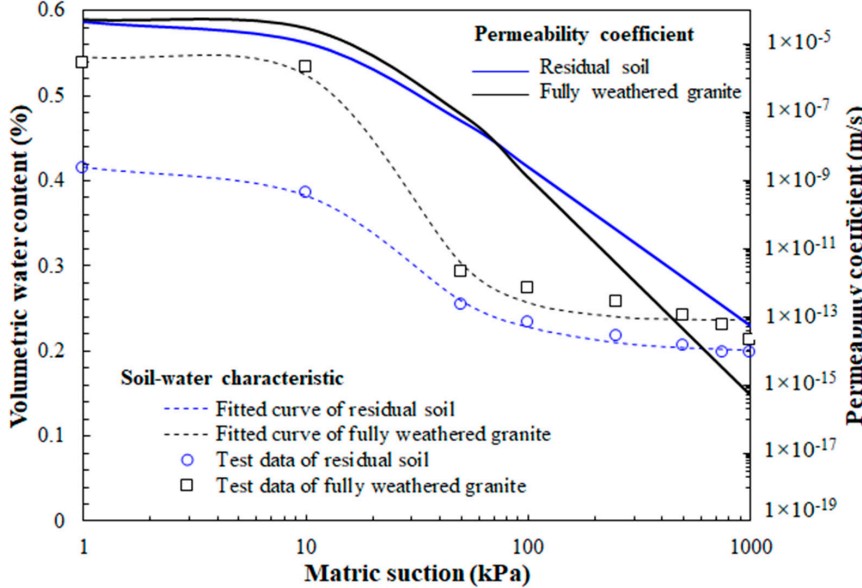

**Figure 7.** Soil water characteristic curves and permeability coefficient of residual soil and fully weathered granite.

**Table 4.** Parameters of SWCCs for residual soil and fully weathered granite.

| Soil Layer | $\theta_r$ | $\theta_s$ | $\alpha$ | $m$ | $R^2$ |
|---|---|---|---|---|---|
| Residual soil | 0.1992 | 0.4169 | 0.0629 | 0.5184 | 0.9975 |
| Fully weathered granite | 0.2357 | 0.5454 | 0.0460 | 0.6381 | 0.9889 |

*3.2. Field Test*

Slope deformation monitoring at points M1, M2, and M3 was conducted from 20 July 2022 to 30 August 2022 in six stages to obtain the horizontal surface displacement, as shown in Figure 6. The monitoring program and precipitation data are listed in Table 5. The data collection interval was 7 d.

**Table 5.** Displacement monitoring stages and corresponding precipitation data.

| Monitoring Stage | Date | Duration (d) | Accumulated Precipitation (mm) | Maximum Daily Precipitation (mm) | Date of the Maximum Daily Precipitation |
|---|---|---|---|---|---|
| ① | 20 July 2022~26 July 2022 | 7 | 87 | 43.1 | 7/23 |
| ② | 27 July 2022~2 August 2022 | 7 | 142 | 53.4 | 7/29 |
| ③ | 3 August 2022~9 August 2022 | 7 | 214 | 84.3 | 8/6 |
| ④ | 10 August 2022~16 August 2022 | 7 | 431 | 112.6 | 8/13 |
| ⑤ | 17 August 2022~23 August 2022 | 7 | 773 | 187.1 | 8/20 |
| ⑥ | 24 August 2022~30 August 2022 | 7 | 1042 | 195.7 | 8/28 |

Figure 8 shows the daily precipitation data recorded at the precipitation stations. The cumulative precipitation exhibited an increasing trend during the monitoring period. The growth rates of the cumulative precipitation from the second to the sixth stages were 63.2%, 146.0%, 395.4%, 788.5%, and 1097.7%, respectively. The maximum daily precipitation of 195.7 mm occurred on August 28, and the maximum hourly precipitation intensity of 71.8 mm/h occurred at 12:00 on that day.

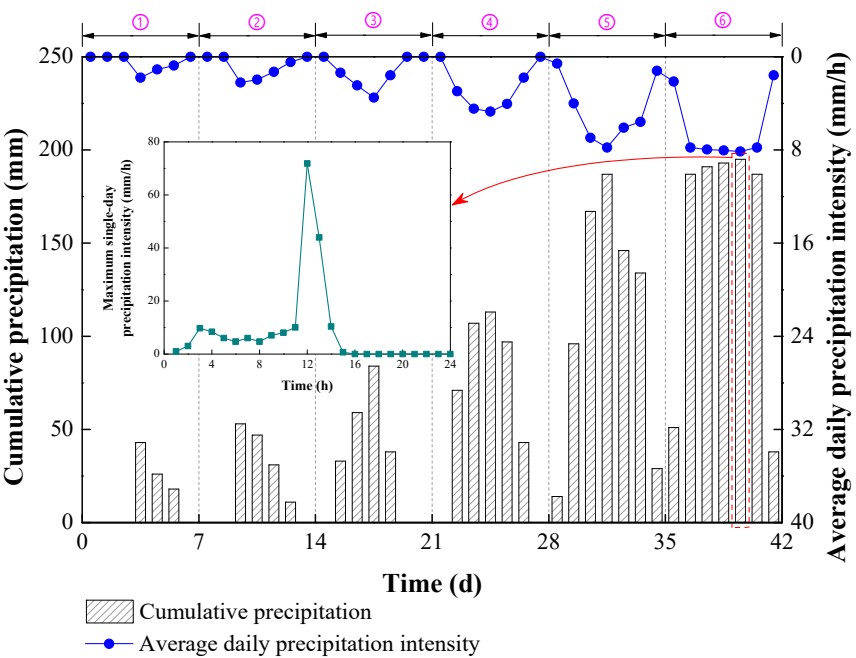

**Figure 8.** Precipitation data during different monitoring stages. (Notes: the number in the figure means the monitoring stage).

Figure 9 shows the temporal variations of the cumulative displacement and cumulative precipitation, indicating a positive correlation between the two parameters. During the first three monitoring stages, the cumulative precipitation was low, and the cumulative horizontal surface displacements were below 20 mm. The precipitation increased by 342 mm from the fourth to the fifth monitoring stages, and the horizontal surface displacement increments of M1, M2, and M3 were 11.7 mm, 12.4 mm, and 17.2 mm, respectively. From the fifth to the sixth monitoring period, the precipitation increased by 269 mm, and the horizontal surface displacement increments of M1, M2, and M3 were 22.3 mm, 26.4 mm, and 31.7 mm, respectively. The results indicate that an increase in the precipitation amount causes a significant increase in the horizontal surface displacement of the cut slope. As shown in Figure 6, M3 is located at the top of the cut slope, M1 is the farthest from the top of the cut slope, and M2 is located between M1 and M3. The ranking of the horizontal surface displacement of the three monitoring points is M3 > M2 > M1. The closer the monitoring point is to the top of the cut slope, the greater the horizontal surface displacement because the excavated soil at the foot of the slope is an unstable free surface. It deformed due to precipitation, resulting in soil displacement at the top of the cut slope.

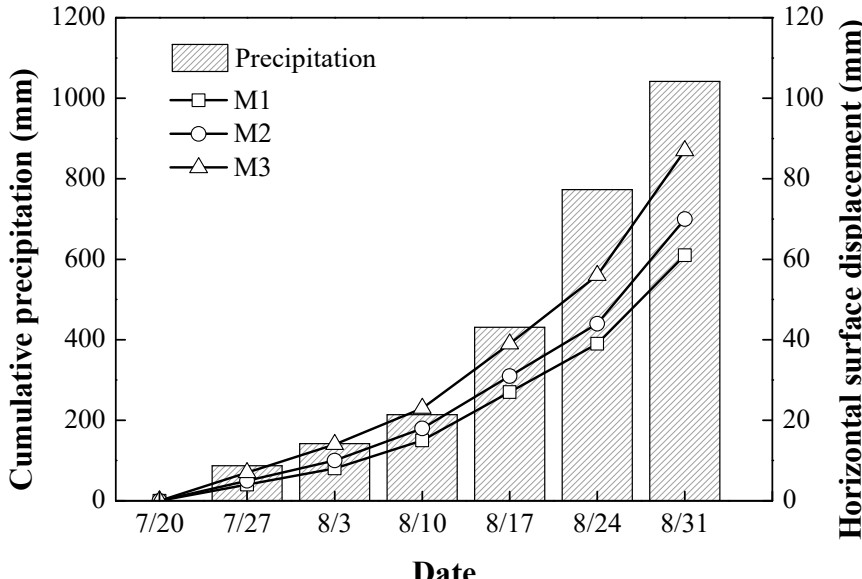

**Figure 9.** Cumulative precipitation and horizontal surface displacement.

*3.3. Numerical Simulation*

The Geo Studio software and the limit equilibrium method were selected for the fluid-solid coupling analysis of the typical slope. The numerical model is shown in Figure 10. In the SIGMA/W module, the deformations of the left and right sides were fixed in the horizontal direction and were allowed to move freely in the vertical direction [31]. In the SEEP/W module, the slope surface had a flow boundary, and the head boundary was less than 475 m on the left and 422 m on the right side. The bottom of the slope had an impervious boundary. The pore water pressure obtained from SEEP/W was imported into the SLOPE/W module for stability analysis. The slope stability was calculated using the Morgenstern–Price method, which is typically used to describe shear damage to soils and rocks [32]. The damage envelope of the model matched the Mohr–Coulomb strength criterion and the tensile damage criterion. The flowchart of the methodology is shown in Figure 11. The details of the analysis are as follows.

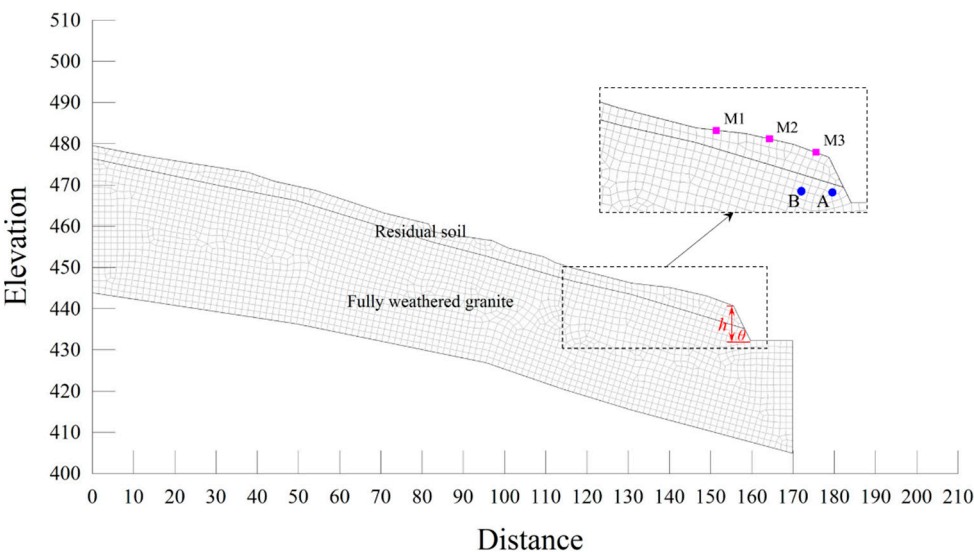

**Figure 10.** Numerical model.

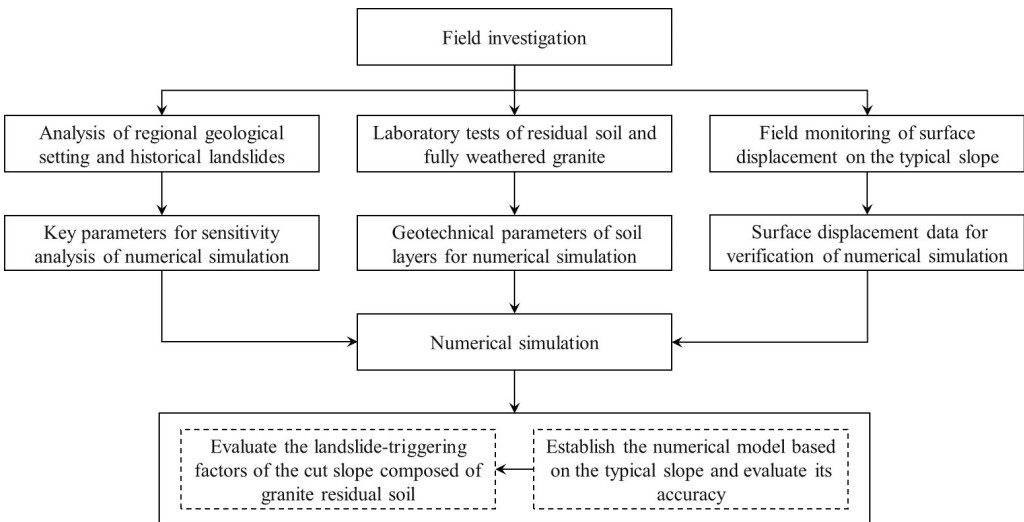

**Figure 11.** Flowchart of the methodology.

A transient unsaturated seepage model coupled with a deformation model was established using the SEEP/W and SIGMA/W modules to evaluate the accuracy of the numerical method. The cumulative precipitation from the first to the sixth stages was simplified into the precipitation duration and intensity (Figure 8), which were applied to the slope surface as the unit flow boundary to enable the coupled seepage and deformation analysis. As shown in Figure 10, the horizontal surface displacement was obtained at M1, M2, and M3 and compared with the monitoring data.

A transient unsaturated seepage coupled with a model to calculate slope stability was established using the SEEP/W and SLOPE/W modules to evaluate the landslide-triggering factors of the cut slope composed of granite residual soil. The different cases of the numerical analysis are listed in Table 6. Different heights at a given gradient (cases H-1 to H-5) and different gradients at a given height (cases G-1 to G-5) were analyzed to assess the seepage and stability characteristics of cut slopes with different parameters. The values of the heights and gradients were based on the historical landslide data (Table 1), and values of a typical slope (8 m and 65°) were included. The hourly precipitation intensity corresponding to the maximum daily precipitation during the monitoring period (28 August 2022) was used as the unit flow boundary in the 10 cases, and the precipitation duration was 7 d.

The precipitation occurred from 1:00 to 15:00 every day, and the remaining period was the non-precipitation phase (from 16:00 to 24:00). In cases P-1 to P-4, four precipitation intensities at a given height and gradient were analyzed. Continuous precipitation occurred for 7 d. The height and gradient were those of the typical slope (8 m and 65°). The precipitation amounts were selected from the classification standard of the precipitation levels posted on the website of the China Meteorological Administration: 35 mm/d (general precipitation), 75 mm/d (heavy precipitation), 150 mm/d (torrential precipitation), and 200 mm/d (extreme precipitation). Two observation points (A and B) were used in all simulations to obtain the pore water pressure trend inside the soil at the foot of the slope, as shown in Figure 10.

**Table 6.** The cases of the numerical simulations.

| Case Number | Gradient of the Cut Slope ($\theta$) (°) | Height of the Cut Slope ($h$) (m) | Precipitation Intensity | Precipitation Duration |
|---|---|---|---|---|
| H-1 | | 2 | | |
| H-2 | | 4 | Monitoring data of precipitation on August 28 | 7 d |
| H-3 | 65 | 6 | | |
| H-4 | | 8 | | |
| H-5 | | 10 | | |
| G-1 | 60 | | | |
| G-2 | 65 | | Monitoring data of precipitation on August 28 | 7 d |
| G-3 | 70 | 8 | | |
| G-4 | 75 | | | |
| G-5 | 80 | | | |
| P-1 | | | 35 mm/d | |
| P-2 | | | 75 mm/d | |
| P-3 | 65 | 8 | 150 mm/d | 7 d |
| P-4 | | | 200 mm/d | |

The stability classification standards derived from the Code for geological investigation of landslide prevention (GB/T 32864-2016) [33] are listed in Table 7.

**Table 7.** Stability classification of the slope.

| Stability Coefficient | $F_s < 1.00$ | $1.00 \leq F_s < 1.05$ | $1.05 \leq F_s < 1.15$ | $F_s \geq 1.15$ |
|---|---|---|---|---|
| State | Highly unstable | Moderately unstable | Moderately stable | Highly stable |

## 4. Results and Analysis

*4.1. Comparison of The Horizontal Surface Displacement Derived from Field Tests and Numerical Simulations*

The simulated and measured horizontal surface displacements at M1, M2, and M3 were compared, and the correlation between them was analyzed to evaluate the numerical model's accuracy. Figure 12 shows the comparison between the measured and simulated horizontal surface displacements. The measured (simulated) surface horizontal displacements at the three points are 14.3, 13.0, and 11.4 times (14.9, 13.8, and 13.2 times) higher in the sixth stage than in the first stage. It is observed that the simulated results are in good agreement with the measured data, and the average relative errors for M1, M2, and M3 are 8%, 5%, and 12%, respectively. The correlation coefficient between the simulated

and measured results is 0.9973 (Figure 13). The good agreement between the simulated results and field measurements indicates the high accuracy and reliability of the proposed numerical model and parameters.

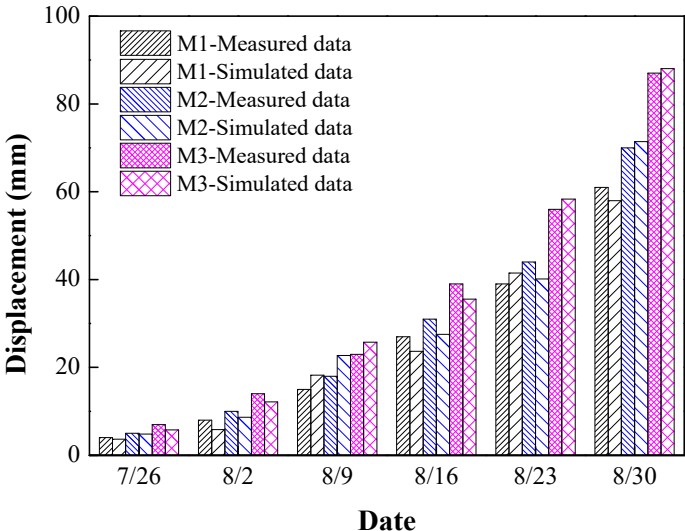

**Figure 12.** Comparison of measured and simulated displacements.

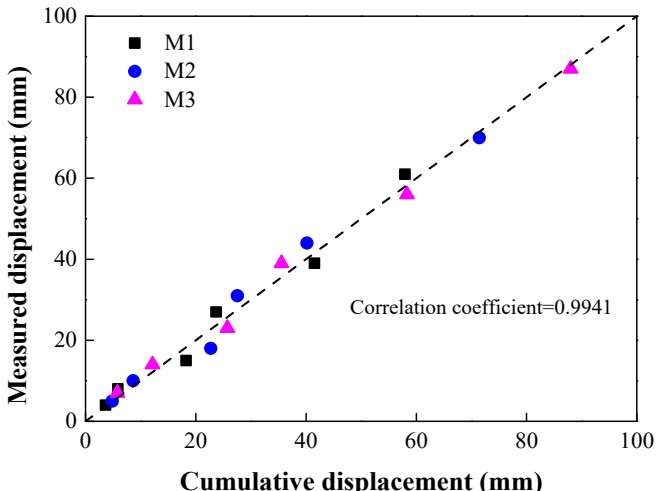

**Figure 13.** Correlation between measured and simulated displacement.

### 4.2. Effect of the Cut Slope's Gradient on Slope Stability

The cut slope has the lowest stability coefficient in the 159th hour. The pore water pressure on the potential landslide surface at this time is shown in Figure 14. At $\theta = 60°$, the horizontal length of the potential sliding surface ($L_s$) ranges from 130.1 to 159.8 m, and the coordinate x corresponding to the positive pore water pressure ($X_p$) ranges from 137.6 to 157.3 m. At $\theta = 80°$, $L_s$ ranges from 130.3 to 157.0 m, and $X_p$ ranges from 138.5 to 155.3 m. Thus, the precipitation infiltration affects the soil saturation in a certain range (coordinate $x$ range: 137–155 m) at the foot of the slope, reducing the matric suction and increasing pore water pressure. $L_s$ and $X_p$ remain unchanged as $\theta$ increases from 60° to 80°.

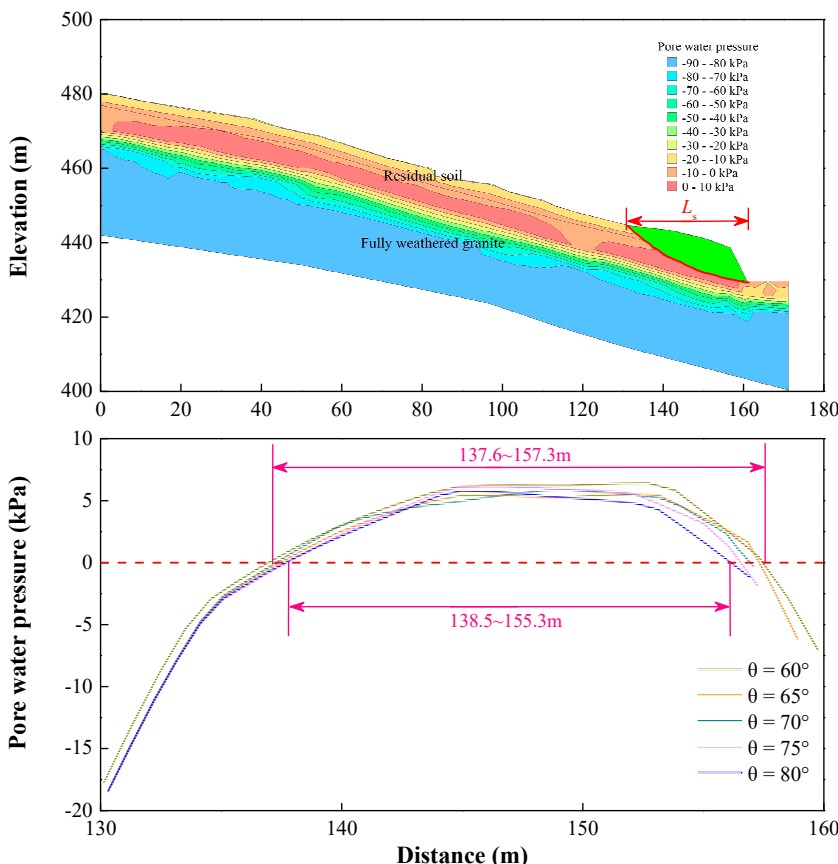

**Figure 14.** Pore water pressure on the potential landslide surface.

The precipitation amount was recorded in two phases: the precipitation phase (1:00–16:00) and the non-precipitation phase (16:00–24:00). The pore water pressure at observation points A and B for different $\theta$ is shown in Figure 15. In one precipitation cycle, the pore water pressure at observation point A was positively correlated with the precipitation intensity. It increased during the precipitation phase and decreased during the non-precipitation phase. In contrast, the pore water pressure at observation point B was not highly correlated with the precipitation intensity. The pore water pressure at observation point A increased and fluctuated during seven precipitation cycles. It increased sharply at the beginning of the first precipitation cycle and reached the maximum in the 63rd hour (the precipitation intensity was the highest at 71.8 mm/h). Subsequently, the pore water pressure decreased during the non-precipitation phase from the 64th to the 72nd hour. The trend of the pore water pressure was similar in the following 96 h. It reached the maximum in the precipitation phase and decreased during the non-precipitation phase. The pore water pressure at observation point B showed an increasing trend, characterized by slow growth from the start to the 63rd hour and a sharp increase in the 64th hour. The reason is that the thickness of the overlying soil is thinner, and the infiltration path of rainwater is shorter at observation point A than at observation point B. Therefore, the response time of the pore water pressure to precipitation is faster at observation point A than at observation point B, and the fluctuation range is large. The pore water pressure response time was 3 h, 3 h, 3 h, 2 h, and 2 h after the start of the precipitation event at point A and 11 h, 11 h, 5 h, 5 h, and 5 h at point B for $\theta$ values of 60°, 65°, 70°, 75°, and 80°. These results indicate that the pore water pressure response time was 1 h and 6 h faster at observation points A and B as $\theta$ increased from 60° to 80°. The reason is that when $\theta$ increases, the overlying soil at the observation point is removed, resulting in a shorter infiltration path of rainwater and a longer response time of the pore water pressure.

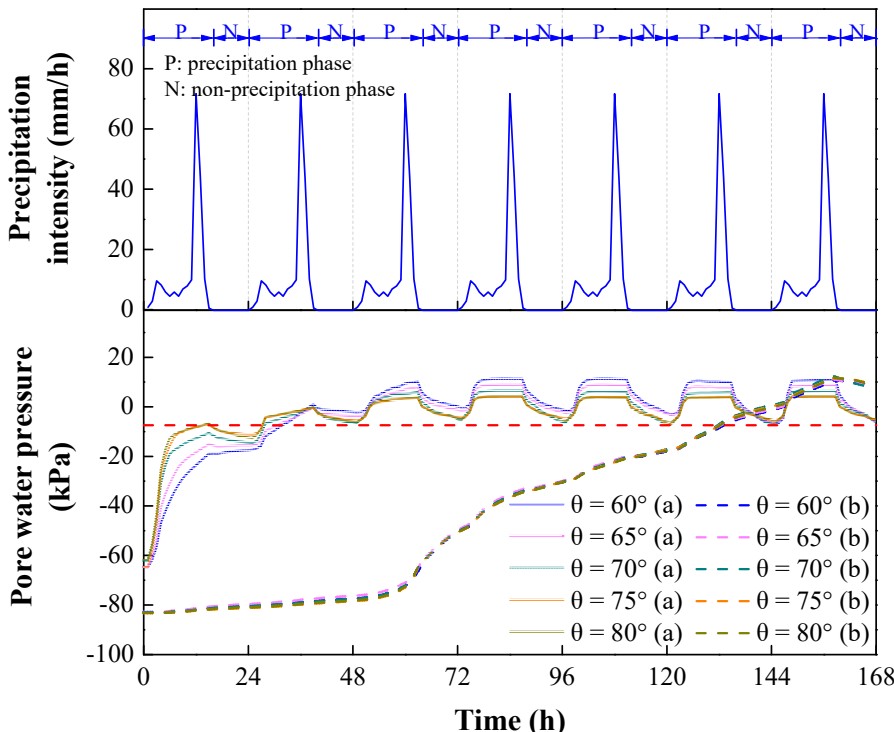

**Figure 15.** Pore water pressure at monitoring points.

The stability coefficients of the cut slope for different $\theta$ are shown in Figure 16. The stability coefficients were positively correlated with the precipitation intensity in each cycle. They decreased in the precipitation phase (1:00–16:00) and increased slightly in the non-precipitation phase (16:00–24:00). The stability coefficients were 1.2%, 2.0%, 2.9%, 4.2%, 7.0%, 9.0%, and 10.4% lower at the end than at the beginning of the seven precipitation cycles. Similarly, the corresponding values were 0.5%, 0.6%, 1.3%, 2.9%, 4.0%, 3.9%, and 5.0% in the non-precipitation phase. The stability coefficients were 16.9% lower at the end of the seventh cycle than at the beginning of the first cycle. The main reason is that the shear strength of the soil decreases during the precipitation phase because the pore water pressure increases due to rainwater infiltration. Therefore, the slope stability decreases. However, during the non-precipitation phase, the pore water pressure dissipates in the shallow soil, and the effective stress rises, increasing the stability coefficient slightly. In addition, when $\theta$ was 60°, 65°, and 70°, the cut slope became moderately unstable in the 106th, 52nd, and 22nd hours at $\theta$ values of 60°, 65°, and 70°, respectively. When $\theta$ was 75° and 80°, the cut slope became moderately unstable before the first precipitation cycle. The rainwater infiltration time is shorter at larger $\theta$ values, and the pore water pressure responds faster. Thus, the slope reached the moderately unstable state faster.

### 4.3. Effect of the Cut Slope's Height on Slope Stability

The cut slope has the lowest stability coefficient in the 159th hour. The pore water pressure on the potential landslide surface at this time is shown in Figure 17. At $h = 2$ m, the horizontal length of the potential sliding surface ($L_s$) ranges from 130.3 to 173.6 m, and the coordinate $x$ corresponding to the positive pore water pressure ($X_p$) ranges from 139.5 to 168.4 m. At $h = 10$ m, $L_s$ ranges from 130.3 to 154.0 m, and $X_p$ ranges from 138.2 to 152.2 m. As $h$ increases from 2 m to 10 m, $L_s$ decreases by about 18.0 m, and $X_p$ decreases by about 15.0 m. The reason is that $h$ increased because some of the soil was removed at the foot of the slope, and $\theta$ remained unchanged. Thus, the horizontal coordinate $x$ on the left side of the potential sliding surface changed slightly, whereas it moved to the left on the right side of the surface. Therefore, $L_s$ decreased with an increase in $h$, and the positive pore water pressure decreased.

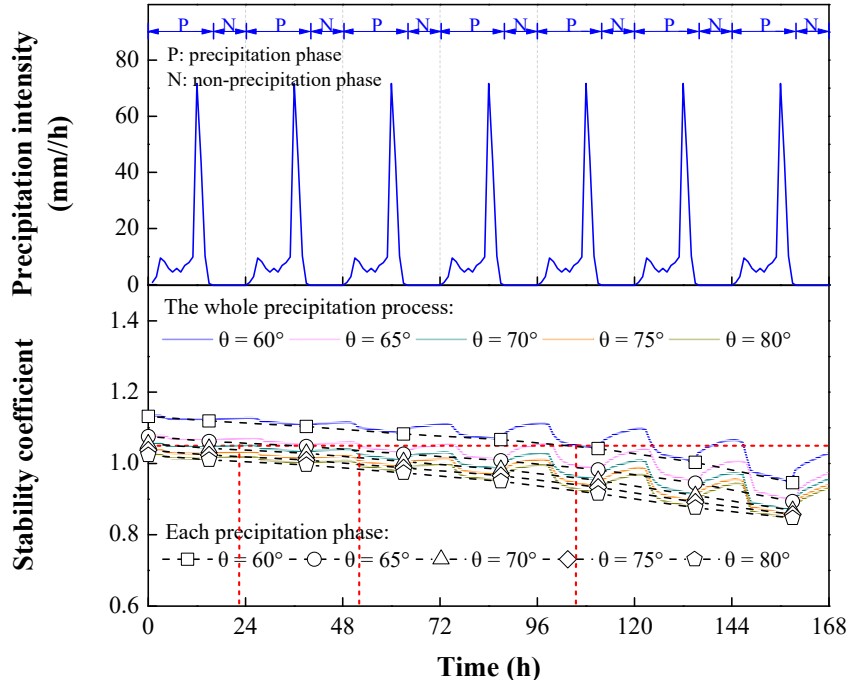

**Figure 16.** Stability coefficients and precipitation intensity.

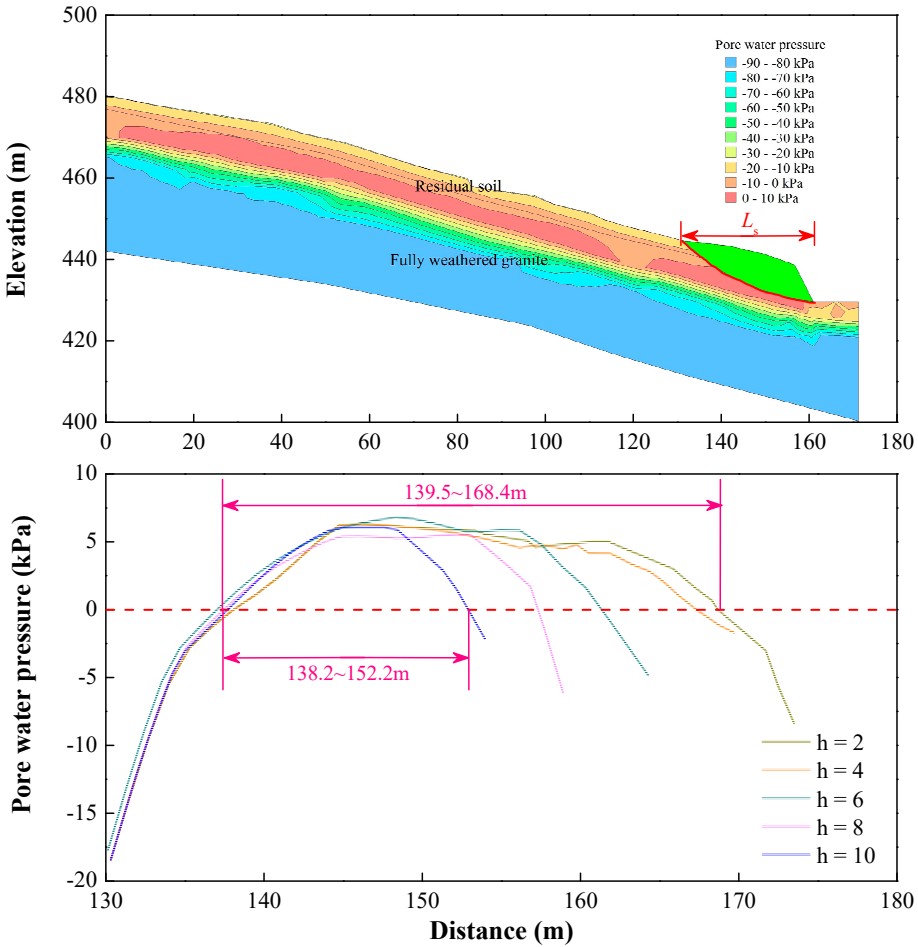

**Figure 17.** Pore water pressure on the potential landslide surface.

The pore water pressure at observation points A and B for different $h$ is shown in Figure 18. The pore water pressure trend at the two observation points is similar to that in Figure 16. The pore water pressure at observation point A was positively correlated with the precipitation intensity at point A but not at observation point B in one precipitation cycle. During seven precipitation cycles, the pore water pressure at observation point A increased sharply to the maximum in the first 63 h and fluctuated in the following 96 h. In contrast, that at observation point B increased slowly in the first 63 h and began to rise sharply in the 64th hour. The pore water pressure response time was 3 h, 3 h, 2 h, 2 h, and 1 h after the start of the precipitation event at point A and 14 h, 14 h, 14 h, 11 h, and 10 h at point B for $h$ values of 2 m, 4 m, 6 m, 8 m, and 10 m, respectively. These results indicate that the pore water pressure response time was 2 h and 4 h faster at observation points A and B as $h$ increased from 2 m to 10 m. As $h$ increases, the soil is removed from the upper right side of the observation point, shortening the rainwater infiltration path, accelerating soil saturation, and increasing the response time of the pore water pressure.

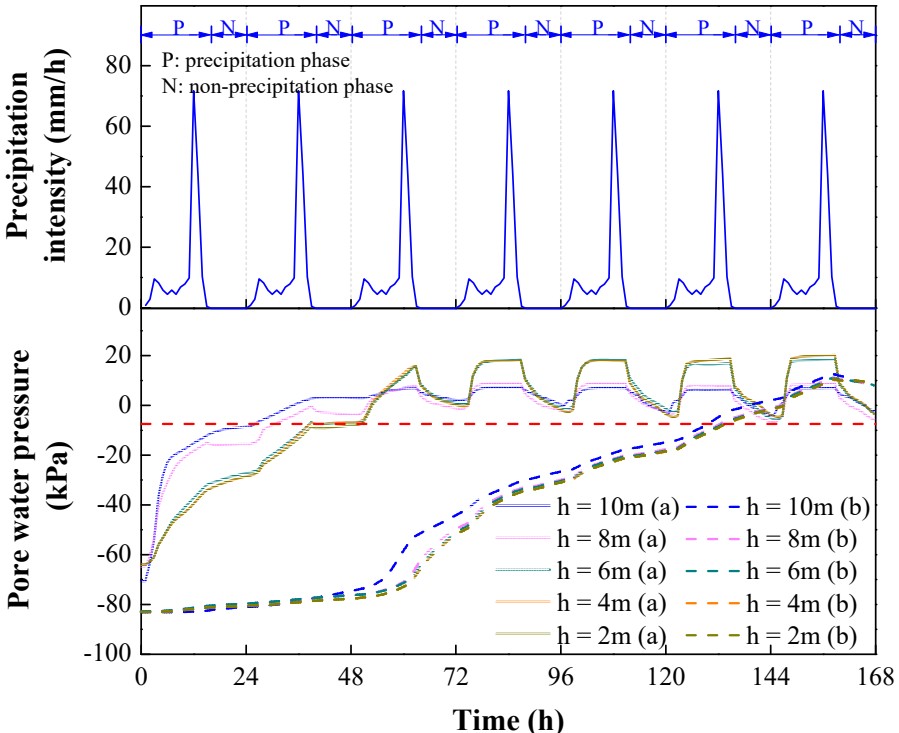

**Figure 18.** Pore water pressure at monitoring points.

The stability coefficients of the cut slope for different $h$ are shown in Figure 19. The stability coefficient decreased in the precipitation phase and rose slightly in the non-precipitation phase. The stability coefficients were 1.3%, 1.8%, 3.2%, 4.7%, 7.3%, 9.3%, and 10.7% lower at the end than at the beginning of the seven precipitation cycles. Similarly, the corresponding values were 0.5%, 0.6%, 1.2%, 2.9%, 4.4%, 4.7%, and 5.9% lower in the non-precipitation phase. As the number of precipitation cycles increased, the stability coefficients decreased. They were 17.8% lower at the end of the seventh precipitation cycle than at the beginning of the first cycle. The rainwater infiltration during the continuous precipitation event increases the soil weight and decreases the cohesion and internal friction angle, reducing the slope's sliding resistance. The rainwater infiltration degree is higher at higher precipitation intensities, resulting in a sharp decrease in the soil's shear strength and slope stability [34]. The stability coefficient of the cut slope in the seven precipitation cycles did not decrease below 1.05 for $h$ values of 2 m, 4 m, and 6 m. When $h$ was 8 m, the cut slope entered a moderately unstable state in the 33rd hour, and when $h$ was 10 m, the cut slope became moderately unstable before the first precipitation cycle. As $h$ increased, the

rainwater infiltration time decreased, and the soil saturation increased, causing the slope to become moderately unstable.

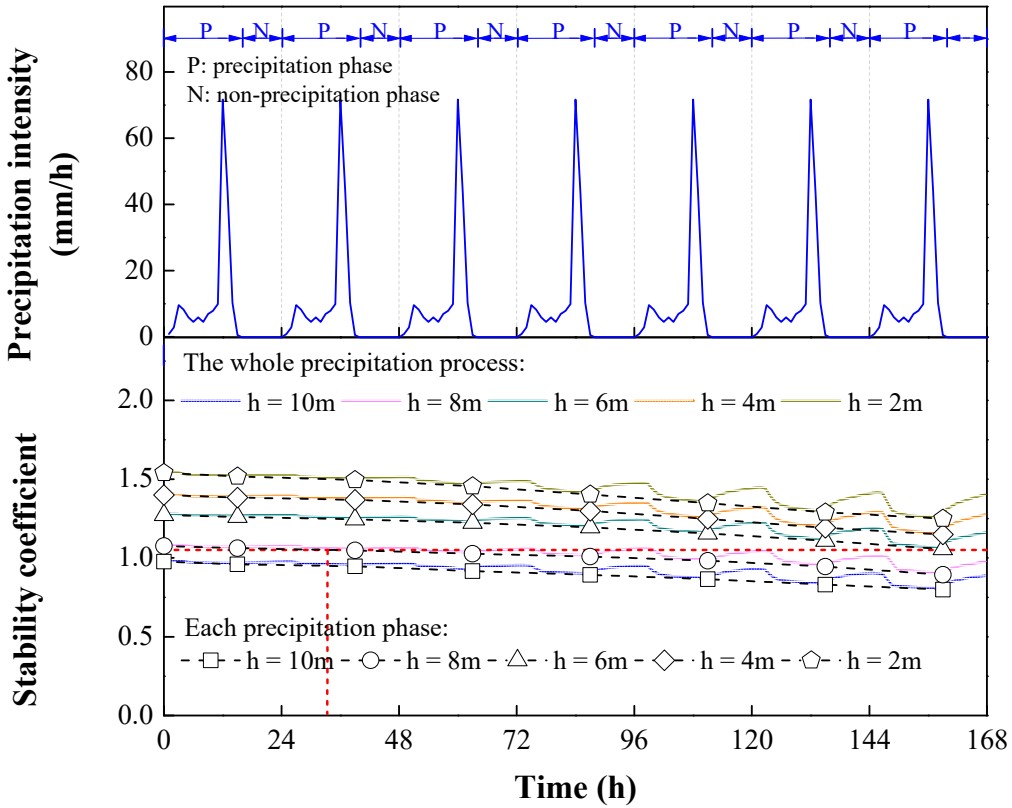

**Figure 19.** Stability coefficients and precipitation intensity.

*4.4. Effect of Precipitation Intensity on Slope Stability*

The pore water pressure at observation points A and B under different precipitation intensities (35 mm/d, 75 mm/d, 150 mm/d, 200 mm/d), $\theta = 65°$, and $h = 8$ m is shown in Figure 20a,b. The pore water pressure at observation point A increased rapidly in the early stage and slowly in the late stage. The pore water pressure reached the maximum value in the 39th hour (the maximum precipitation intensity was 71.8 mm/h). However, the pore water pressure at observation point B exhibited the opposite trend, i.e., slow growth at the beginning and rapid growth at the end. The pore water pressure reached the maximum value in the 135th hour (the maximum precipitation intensity was 71.8 mm/h). The response time was much faster at observation point A than at observation point B because of the lag time of rainwater infiltration [35]. At the start of the precipitation event, the pore water pressure in areas of shallow soil responded quickly, increased rapidly, and then leveled off. The higher the precipitation intensity, the faster the pore water pressure increased and the earlier it stabilized. In addition, at low precipitation intensities, the rainwater infiltrated in the radial direction, whereas the vertical infiltration intensity and volume were low. Thus, the pore water pressure in the deep soil did not increase significantly. As the precipitation intensity increased, the radial and vertical infiltration intensities rose, and the internal pore water pressure of the deep soil increased [1].

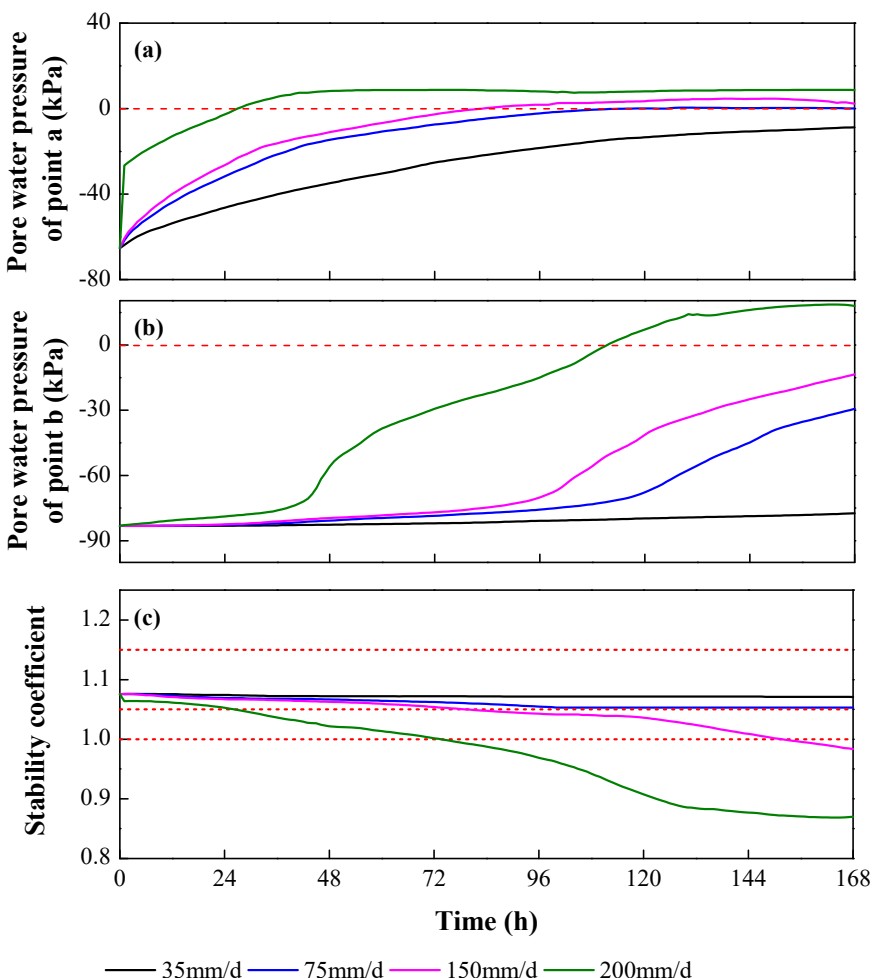

**Figure 20.** Pore water pressure and stability coefficient for different precipitation intensities: (**a**) pore water pressure of monitoring point A; (**b**) pore water pressure of monitoring point B; (**c**) stability coefficients.

The trends of the slope stability coefficients under four rainfall intensities (35 mm/d, 75 mm/d, 150 mm/d, 200 mm/d), $\theta = 65°$, and $h = 8$ m are shown in Figure 20c. The slope stability decreased during the precipitation cycles and with an increase in the precipitation intensity. The cut slope reached the unstable state faster. The stability coefficients for the four precipitation intensities were 0.4%, 3.7%, 8.6%, and 19.1% lower at the end than at the beginning of the seven precipitation cycles. At precipitation intensities of 35 mm/d and 75 mm/d, the stability coefficient decreased slowly, and the cut slope remained stable at the end of the seventh precipitation cycle. At a precipitation intensity of 150 mm/d, the stability coefficient of the slope decreased slowly from the beginning of the first to the end of the fifth precipitation cycle. The subsequent rate of decrease was faster from the beginning of the sixth to the end of the seventh precipitation cycle. At the end of the seventh cycle, the stability coefficient was lower than 1.0, and the cut slope was in an unstable state. At a precipitation intensity of 200 mm/d, the slope stability coefficient decreased slowly during the first precipitation cycle, and the decreasing trend accelerated at the beginning of the second precipitation cycle. The stability coefficient was lower than 1.0 at the end of the third precipitation cycle, and the cut slope was highly unstable. These results indicate that long-term torrential precipitation or short-term extraordinary precipitation can trigger landslides of cut slopes. The reason is seepage occurs in the shallow layers of the slope during short-term extraordinary precipitation events. Thus, the pore water pressure of the soil at the foot of the slope changes rapidly, potentially triggering a shallow landslide [36]. However, under long-term torrential precipitation, the rainwater continues to infiltrate into

the interior of the slope. The shallow soil saturates first, and the water migrates to the foot of the slope due to gravity, resulting in a temporary saturation zone at the foot of the slope. The shear strength of the soil decreases, triggering a deep-seated landslide [37,38].

## 5. Discussions of the Landslide Mechanism

According to the field investigation and numerical simulation results, the formation and evolution of the cut slope landslide can be divided into the following four stages:

(a) Slope formation and evolution. A slope composed of granite residual soil was formed due to the climate, tectonics, and physical and mechanical properties of rock and soil. This soil is highly permeable, and its strength decreases rapidly as the water content increases.

(b) Formation of unloading zone at slope foot. As shown in Figure 21, cutting the slope disrupted its mechanical equilibrium, forming an unloading zone. When the leading edge was cut, the stress of the unloading zone was released, and tensile fissures occurred. However, the slope remained stable.

(c) Migration and loss of soil particles. The sand particles in the granite residual soil act as the skeleton and the clay and silt particles are attached to the skeleton, forming a combined structure [38] (Figure 21). Since the soil is a porous medium, the seepage field affects the soil skeleton due to rainwater infiltration. The change in the pore water pressure affects the effective stress on the soil skeleton; thus, the soil skeleton is deformed, and its strength is reduced.

(d) Instability of cut slope. Due to rainfall infiltration, the pore water pressure in the unsaturated soil rises, the matric suction decreases, and the effective stress decreases. As a result, the shear strength of the soil and the slope stability decrease. In addition, tensile fissures occur at the top of the cut slope due to gravity and seepage, resulting in a landslide.

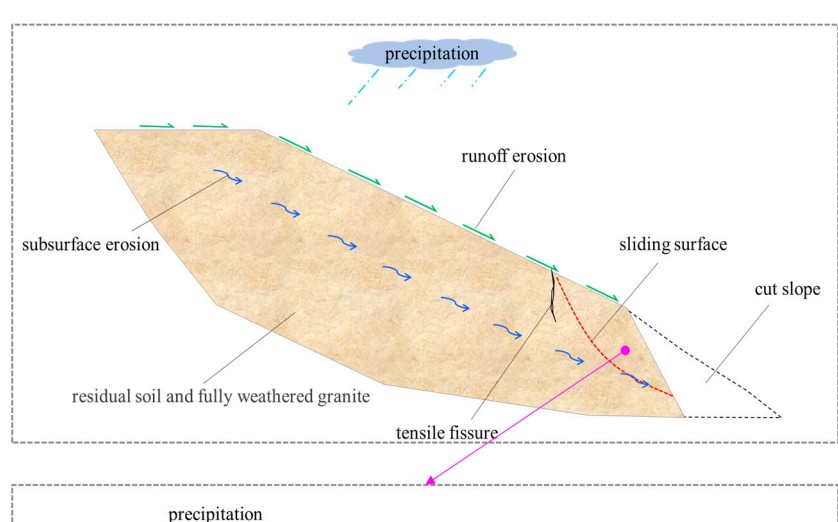

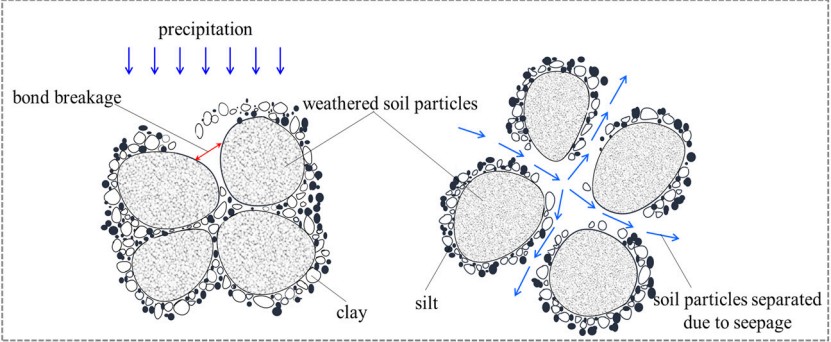

**Figure 21.** Schematic diagram of the formation and evolution of the landslide on the cut slope composed of granite residual soil.

## 6. Engineering Implications

The following suggestions are provided to prevent landslides of cut slopes composed of granite residual soil in southwest Zhejiang.

(1) Before the start of the engineering project, a detailed engineering and geological investigation should be carried out on the slope to determine the soil thickness and distribution and prevent construction in unfavorable geological areas such as fault zones.

(2) The critical height values of cut slopes are 5.3 m, 5.5 m, 5.7 m, 6.0 m, and 6.3 m for slope gradients of 60°, 65°, 70°, 75°, and 80°, respectively, as shown in Figure 22. If these values are exceeded, the slope may become unstable. In addition, the cut area should be strengthened early during construction, using a concrete retaining wall, dry blocks, mortar blocks, or other methods.

(3) After the completion of the engineering project, precipitation and displacement monitoring equipment should be installed on the top of the cut slope. The stability coefficient is lower than 1.0 during long-term torrential or short-term extraordinary precipitation events with durations of 26 h and 78 h, as shown in Figure 23. A landslide of the cut slope can occur under these conditions. Therefore, the displacement at the foot of the slope should be monitored, and if necessary, people in danger areas should be evacuated immediately to prevent loss of life and property.

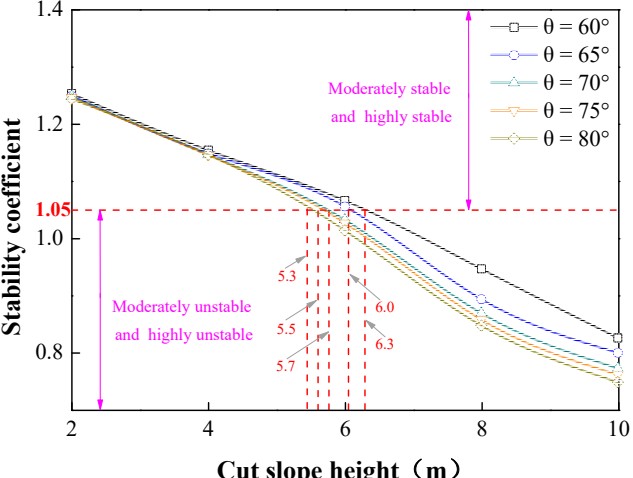

**Figure 22.** Relationship between the stability coefficient, $\theta$, and $h$.

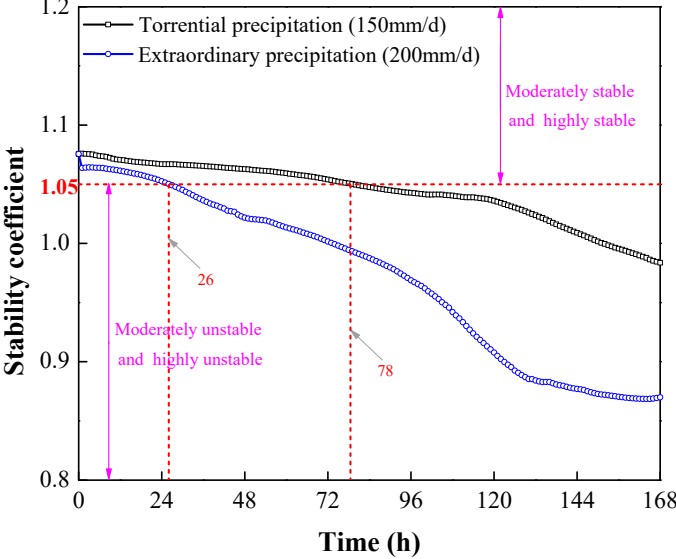

**Figure 23.** Relationship between the stability coefficient and precipitation intensity.

## 7. Conclusions

The characteristics of historical landslides in Chongtou Town in southwestern Zhejiang were summarized, and a typical slope was selected for analysis. The hydraulic and mechanical properties of the residual soil and fully weathered granite were tested, and the surface displacements of the slope were monitored. Geo-studio was used to establish a coupled seepage-deformation model to validate the numerical method and investigate the landslide-triggering factors of the cut slope. The main conclusions are as follows:

(1) All historical landslides in the study area were small-scale and occurred at the foot of the slope. Different precipitation intensities and durations were observed 168 h before the landslide occurred. The intensity was general or heavy during this period but was the highest (heavy or torrential) on the date of the landslide.

(2) The initial dry density, unit weight, shear strength, Poisson's ratio, and saturated permeability coefficient of the residual soil and fully weathered granite were similar. The deformation modulus was 2.3 times larger for the fully weathered granite than the residual soil.

(3) The field monitoring results showed that the deformation of the cut slope was positively correlated with the cumulative precipitation. The simulated and measured results were in good agreement, indicating that the proposed numerical model and parameters were accurate and reasonable.

(4) As $\theta$ or $h$ of the cut slope increased, the stability coefficient decreased, the response time of the pore water pressure at the observation points increased, and the horizontal length of the potential sliding surface decreased. The critical values of $h$ were 5.3 m, 5.5 m, 5.7 m, 6.0 m, and 6.3 m at $\theta$ values of 60°, 65°, 70°, 75°, and 80°, respectively.

(5) Long-term torrential precipitation or short-term extraordinary precipitation can trigger landslides of cut slopes. The stability coefficient was lower than 1.0 during precipitation events with durations of 26 h and 78 h, with a high likelihood of landslides of cut slopes.

(6) The landslide causes of the cut slope composed of granite residual soil in southwest Zhejiang can be attributed to internal and external factors. The internal factors include the geotechnical soil properties and the slope's structure, and the dominant external factor is precipitation. The formation of the slope and landslide includes four stages: slope evolution, formation of an unloading zone at the slope foot, migration and loss of soil particles, and instability of the cut slope.

This study focused on the landslide-triggering factors of a cut slope composed of granite residual soil. Future studies will investigate the instability mechanism of the slope by monitoring the temporal and spatial variations of multiple indicators in the field and employing multiple theories for stability analysis.

**Author Contributions:** Writing original draft, validation, and formal analysis, T.Y.; review and editing, J.X.; data curation, L.Y.; investigation, J.G.; conceptualization, methodology, and supervision, H.X. All authors have read and agreed to the published version of the manuscript.

**Funding:** This research was funded by the National Natural Science Foundation of China (Grant No. 42172309).

**Institutional Review Board Statement:** Not applicable.

**Informed Consent Statement:** Not applicable.

**Data Availability Statement:** The data presented in this study are available on per request to the corresponding author.

**Acknowledgments:** The authors want to express their deep thanks to the anonymous reviewers for their constructive comments.

**Conflicts of Interest:** The authors declare no conflict of interest.

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
