# Peer review of "Field Investigation and Finite Element Analysis of Landslide-Triggering Factors of a Cut Slope Composed of Granite Residual Soil: A Case Study of Chongtou Town, Lishui City, China"

_sustainability, doi:10.3390/su15086999_

Round 1

Reviewer 1 Report

This study summarizes the findings of the investigation of the effects of heavy precipitation on the hydrological response, stability, and evolutionary mechanisms of cut slopes in granite soil regions. It is useful as a case study because it reports the collapse of a cut slope due to rain in an area where granite residual soil is distributed and the results of parametric numerical analysis. Another characteristic is that the period covered by the study of rainfall is relatively long. The parameter study by numerical analysis is also well written.

   Comments of the content

1)    Page 3, the last line: Term “torrential precipitation” should be defined clearly.

2)    Page 4, Figure 2: Please indicate the number of failure slope corresponding to a), b) and c), respectively.

3)    Table1: If possible, Thickness of the residual soil and fully weathered granite layer should be indicated in table 1.

4)    In addition to above, please indicate how to judge the residual soil part and fully weathered part.

5)    Section 2.3: Show data similar to Table 1 for selected slopes.

6)    Page , Section 2.3: Please show what kind of monitoring devise were used.

7)    Figure 8: If possible, the difference among M1, M2 and M3 should be commented.

8)    In addition to above, on the discussion of the simulation results, it is better to indicate the comments on the difference among M1, M2 and M3. 

Author Response

The authors would like to express their appreciation to the reviewers for their constructive comments, which are of great help in improving the manuscript. Details of the revisions and point-by-point responses are given in the following. All the modifications were marked in red color in the revised manuscript.

Reviewer 2 Report

The comments can be found in the file herewith attached.

Author Response

(The authors gave the same response as above.)

Round 2

Reviewer 2 Report

No comments
